# Emergency, ABO-Incompatible Living Donor Liver Re-Transplantation for Graft Failure Complicated by Pneumonia-Associated Sepsis

**DOI:** 10.3390/jcm12031110

**Published:** 2023-01-31

**Authors:** Seoung Hoon Kim, Young-Kyu Kim

**Affiliations:** 1Organ Transplantation Center, National Cancer Center, 323 Ilsan-ro, Ilsandong-gu, Goyang-si 10408, Republic of Korea; 2Department of Surgery, Jeju National University School of Medicine, Aran 13gil 15 (Ara-1Dong), Jeju-si 63241, Republic of Korea

**Keywords:** ABO-incompatible, re-transplantation, graft failure, living donor liver transplantation, chronic rejection

## Abstract

Although liver re-transplantation is the only therapeutic option for acute and chronic graft failure, few studies have addressed the use of ABO-incompatible living donors in the emergency setting. Here, based on our experience, we report a successful case of emergency, ABO-incompatible, adult-to-adult, living donor liver re-transplantation (LDLT) for late graft failure from chronic rejection complicated by pneumonia-related sepsis. A fifty-five-year-old man had undergone LDLT for hepatocellular carcinoma accompanied by hepatitis C virus (HCV)-related cirrhosis in 30 September 2013. The voluntary donor was his 56-year-old wife, who was also a carrier of HCV. The donor and recipient blood types were the same: O and Rh positive. She underwent a right hepatectomy and was discharged on postoperative day (POD) seven. The patient was also discharged without complications on POD eleven and was followed up with on an outpatient basis. Abdominal distension and jaundice were developed at 6 months after LDLT, when the serum total bilirubin level was 2.7 mg/dL. The serum total bilirubin levels increased rapidly to 22.9 mg/dL over the next 4 months. Chronic rejection was diagnosed via liver biopsy. On 3 October 2014, he developed pneumonia-related sepsis and showed the progressive deterioration of liver function. Liver re-transplantation using the right liver from his ABO-incompatible, 20-year-old nephew was performed as an emergency in 15 October 2014. The donor blood type was A and Rh positive. The resection of the failed graft and the implantation of a new graft was performed by the intragraft dissection technique to re-use previously transplanted graft vessels in order to cope with severe adhesions. The recipient went through a gradual recovery process and was finally discharged on POD 50 with normal liver function, while the donor had an uneventful recovery and was discharged on POD 7. Biloma due to bile leak was detected three months after re-transplantation and was cured by percutaneous interventional procedures. Since then, the postoperative course has been event-free at regular outpatient follow-ups. The patient has so far had normal laboratory findings and no signs of complications. It has been 98 months since the re-transplantation, and the recipient and two donors are still in good condition with normal liver function, having complete satisfaction with the results obtained from this re-transplantation. In conclusion, long-term, satisfactory outcomes can be achieved in emergency, ABO-incompatible, adult-to-adult, living donor liver re-transplantation for graft failure complicated by pneumonia-related sepsis in selected patients.

## 1. Introduction

Living donor liver transplantation (LDLT) is currently the most common form of liver transplantation in Asia. When compatible living donor is unavailable, ABO-incompatible (ABO-I) LDLT can become the ultimate therapeutic choice. Recently, ABO-I LDLT has been reported to have acceptable outcomes in the era of rituximab-based prophylaxis [1,2], and has been accordingly performed while expanding the donor pool.

Re-transplantation using a living-donor graft is also a last resort for a deteriorating patient with irreversible graft failure, especially in the regions with limited access to deceased donors in the current era of LDLTs performed worldwide. However, liver re-transplants are technically complex and difficult. Although improved outcomes have been shown for graft and patient survival rates in trends over time [3,4], re-transplant outcomes remain significantly worse than those for primary transplants [5].

The use of ABO-I liver graft in re-transplantation setting has never been reported in the setting of emergency and sepsis. Here, we introduce our experience with an emergency case of ABO-incompatible, adult-to-adult, living-donor liver re-transplantation for graft failure complicated by pneumonia-related sepsis, and discuss the surgical indications and operative technical difficulties of re-transplantation.

## 2. Case Presentation

Afifty-five-year-old man was referred to the outpatient clinic of the authors’ institution for the evaluation of LDLT for hepatocellular carcinoma and hepatitis C virus (HCV)-related liver cirrhosis on 17 September 2013. He had previously been treated with radiofrequency ablation and transarterial chemoembolization for hepatocellular carcinoma at another hospital. The treatment regimens, using pegylated interferons in combination with ribavirin, had also been used to treat the HCV infection. The patient had no history of smoking or drinking, and no family history of cancers. His liver function was not severely impaired (Child–Pugh B class and a model for end-stage liver disease (MELD) score of 14). The platelet count was reduced to 34/μL. The serum level of HCV RNA, determined by real-time polymerase chain reaction, was less than 12 IU/mL (1.08 Log IU/mL), below the limit of quantitation, which was classified as HCV RNA negative. The serum alpha-fetoprotein (AFP) level was 284.2 ng/mL. His height was 167 cm and his weight was 73 kg, with a body mass index of 26.2. His body surface area was calculated to be 1.82 m^2^ according to DuBois and DuBois [6].

After a full work up, his fifty-six-year-old wife voluntarily donated her right liver graft. Her height was 161 cm and her weight was 52 kg, with a body mass index of 20.1. She was a carrier of HCV with normal liver function, but her HCV RNA was negative. A preoperative liver biopsy showed mild centrilobular congestion with no evidence of chronic active hepatitis.

The donor and recipient blood types were the same, O and Rh positive, and the LDLTs were approved by KONOS (Korean Network for Organ Sharing).

The LDLT was conducted on 30 September 2013. All the main procedures were performed by a single surgeon (S.H.K) [7]. Donor surgery was performed as previously reported [8]. The intraoperative wedge biopsy specimen revealed 5% macrovesicular steatosis. The oeration time was 135 min with minimal blood loss. The real weight of the resected right liver was 527 g. The real GRWR was 0.72, which implied a small-for-size graft. The donor was discharged with normal liver function on the seventh day after surgery.

The recipient’s liver transplant went smoothly. The cold ischemia time was 111 min; the warm ischemia time was 24 min; and the anhepatic period was 135 min. The operation time was 6 h. The blood loss was 2000 mL. He was discharged with no complications on postoperative day. The explanted whole liver had a hepatocellular carcinoma outside the Milan criteria (seven nodules; the largest was 3.5 cm in diameter) without angioinvasion (Figure 1), and the histopathological diagnosis of nodules was classified as T2 by American Joint Committee on Cancer (AJCC) 2010 staging. The patient was followed up with at an outpatient clinic once a month. As immunosuppression, a tacrolimus trough level was maintained at approximately 10 ng/mL, with mycophenolate mofetil added.

The patient showed normal liver function for up to 6 months after the LDLT. At 3 months after the LDLT, the serum total bilirubin level, AST level, and ALT level were 0.6 mg/dL, 20 U/L, and 18 U/L, respectively. At 5 months after LDLT, the serum total bilirubin level, AST level, and ALT level were 0.4 mg/dL, 32 U/L, and 14 U/L, respectively. At 6 months after LDLT, the patient complained of abdominal distension. The serum total bilirubin level was 2.7 mg/dL, and the serum level of HCV RNA had increased to 6,890,807 IU/mL. An Abdominal Computed Tomography (CT) revealed patent hepatic vessels, and a minimal IHD dilatation, but a large amount of ascites. Pegylated interferons was started in combination with ribavirin. Ascitic tap procedures were carried out, with diuretics prescribed. However, the abdominal distension and jaundice were aggravated, so that he was admitted for further treatment at 10 months after the LDLT. To rule out the possibility of hyperbilirubinemia due to biliary stenosis, an endoscopic insertion of a retrograde biliary drainage tube was performed.

Despite such treatment, the serum total bilirubin level increased dramatically, up to 22.9 mg/dL. A liver biopsy was performed, and the pathologic findings showed biliary ductopenia suggestive of chronic rejection (Figure 2). Steroid pulse therapy and increased tacrolimus doses did not resolve the rejection. A humoral component of rejection could not be fully excluded, and so rituximab was administered as a rescue therapy in a single dose of 600 mg (375 mg/body surface area), which corresponding to 60 days prior to re-transplantation. All attempted treatments were ineffective.

On 3 October 2014, the serum total bilirubin was 23.0 mg/dL. The abdominal CT scan revealed a parenchymal hematoma in the graft, with a stent visible in the bile duct and abundant ascites in the peritoneal cavity (Figure 3a). Additionally, the patient developed pneumonia (Figure 4a). The heart rate reached a maximum of 130 beats per minute and the body temperature rose to 38.0 °C. The respiration rate was over 30 breaths per min and the blood oxygen saturation dropped to 88%. His blood pressure dropped to 89/57 mm of mercury (mmHg). His mental status began to deteriorate. All the findings indicated shock due to sepsis. He was therefore transferred to the intensive care unit, intubated, and mechanically ventilated. Vasopressors were administered to treat low blood pressure. The cytomegalovirus (CMV) antigenemia assay showed positive results, so a preemptive intravenous injection of ganciclovir was initiated. Klebsiella pneumoniae was identified in the sputum culture and endotracheal aspirate culture, and antibiotics were initiated to treat pneumonia and sepsis. Fortunately, hemodialysis was not required; his renal function remained normal. His MELD score was 23, with a serum total bilirubin level of 29.1 mg/dL. In the state of liver failure, it was judged that pneumonia and sepsis were difficult to improve, so liver re-transplantation was seriously considered.

At the request of a preoperative risk evaluation, both the cardiologist and pulmonologist expressed skepticism regarding re-transplantation. Echocardiography revealed an ejection fraction of 41% and a mild systolic dysfunction of the dilated left ventricle. Successive chest X-rays showed worsening opacity on both sides of the lungs, suggesting a progressively increasing pattern of pulmonary edema. The ratio (P/F ratio) of arterial blood oxygen partial pressure (PaO_2_) to inhaled oxygen fraction (FiO_2_) was 93.5 mmHg, which was diagnosed as acute respiratory distress syndrome. However, blood pressure was stabilized and maintained above 100/60 mmHg even without vasopressors.

The two doctors were concerned that the risk of brain damage during the period of intraoperative hypoxia could be significant, and that the increased intraoperative fluid load and lung burden from blood transfusions could worsen the patient’s condition and even lead to death. There were also concerns about the possibility of continued respiratory failure after surgery and the possibility of prolonged treatment in the intensive care unit. Despite being informed of all the risks, the family desperately wanted a liver re-transplant. His twenty-year-old nephew volunteered to donate his right liver. His height was 176 cm and his weight was 72 kg, with a body mass index of 23.2. His blood type was A and Rh positive. The calculated MELD score just before re-transplantation was 26, with a serum total bilirubin level of 30.7 mg/dL. Arterial blood gas analysis just prior to re-transplantation showed a pH of 7.26, PaCO_2_ of 34.6 mmHg, PaO_2_ of 57.2 mmHg, and an oxygen saturation of 82.9%. A chest X-ray obtained on the morning of the re-transplant day showed extensive consolidation involving both lungs diffusely (Figure 4b).

On 15 October 2014, the ABO-incompatible, living donor liver re-transplantation was performed. The donor hepatectomy was performed with the same technique previously described [8]. The intraoperative donor-liver wedge biopsy specimen revealed no steatosis. The operation time was 154 min, with a blood loss of 300 mL. The real weight of the resected right liver was 630 g and the real GRWR was 0.95. The patient was discharged safely and soundly without complications on POD 7.

The recipient’s liver transplant surgery was very tough, with severe adhesion and easy touch bleeding. In order to cope with the difficulties, the resection of the failed graft and the implantation of a new graft was performed by the intragraft dissection technique to re-use previously transplanted graft vessels [9]. The explanted right liver allograft had green bile staining throughout the capsule and parenchyma, with an intraparenchymal organizing hematoma measuring 4.0 × 4.0 × 3.5 cm (Figure 5).

The cold ischemia time was 75 min, the warm ischemia time was 15 min, and the anhepatic period was 90 min. The peration time was 6 h and 25 min. Blood loss was 12,000 mL.

After retransplant surgery, the patient was returned to the intensive care unit and continued mechanical ventilation while he was intubated. Ventilator weaning started on POD 9, and extubation was performed on POD 14. He was transferred to a general ward on POD 30. The serial chest X-ray findings showed improvement with an almost complete resolution of the abnormalities (Figure 4).

After re-transplant, the patient recovered and demonstrated improving laboratory findings with time. On the day before the LDLT, the AST level was 99 U/L and the ALT level was 32 U/L. On postoperative day (POD) 1, these levels were 153 U/L and 171 U/L, respectively; on POD 7, 35 U/L and 123 U/L, respectively; on POD 14, 39 U/L and 37 U/L, respectively; and on POD 50 at discharge, 19 U/L and 10 U/L, respectively.

The serum total bilirubin level that had been 30.7 mg/dL just before the LDLT was decreased to 13.6 mg/dL on POD 1, but gradually decreased to 8.7 mg/dL on POD 7, 6.7 mg/dL on POD 14, 3.5 mg/dL on POD 21, and was finally 1.5 mg/dL on POD 50, at discharge.

The international normalized ratio was 1.49 before the LDLT, but it increased to 2.04 on POD 2. The ratio was 1.86 on POD 3 and 1.53 on POD 7. The ratio recovered to 1.18 on POD 14.

The isoagglutinin titer was 1:4 before re-transplant, 1:2 on the day of surgery, and below the measurement limit thereafter.

The recipient was discharged from the hospital without complications and with normal liver function on POD 50. The total number of hospitalization days for surgery was 98 days, 48 days before surgery, and 50 days after surgery. The total number of days in the intensive care unit for surgery was 42 days, 12 days before surgery, and 30 days after surgery. The total number of days on mechanical ventilation for surgery was 26 days, 12 days before surgery, and 14 days after surgery.

During the outpatient follow-up, biloma was detected 3 months after surgery (Figure 3d), and percutaneous catheter drainage and transhepatic biliary drainage were performed. The tubes were removed 10 months after surgery. Since then, the postoperative course has been event-free at regular outpatient follow-ups. The patient has, so far, had normal laboratory findings and no signs of complications.

A series of follow-up liver CT scans showed improved results with sufficient liver regeneration in the recipient (Figure 3).

It has been 98 months since the re-transplantation, and both the donor and the recipient are still in good conditions with normal liver functions, having complete satisfaction with the results obtained from this re-transplantation. The patient’s wife, who was the donor at the time of the first liver transplant, is also doing well now, 111 months after the donation operation.

## 3. Discussion

This is the first reported case of an ABO-I LDLT for re-transplantation. Good long-term, clinical results were obtained, despite surgery having been performed in emergency and septic situations. In fact, re-transplantation was not recommended at that time, as the patient was considered too critical for surgery. There were many doubts about the legitimacy of the surgery, not to mention the results of the surgery, but at this point, looking back on the past and considering the current good results, it can humbly be said that the re-transplantation was the best option to save the patient.

Both the donor and recipient were HCV positive at the time of the first LDLT. Between 1995 and 2016, 4.1% of deceased donors in the United States were HCV seropositive [10]. While outcome data for the transplantation of HCV-positive livers to HCV-negative recipients remain scarce, HCV-positive recipients of HCV-positive livers have demonstrated acceptable patient and graft survivals up to 5 years post-transplant [11]. The recipient was treated for HCV reinfection at 6 months after the first LDLT; this may have played a role in causing the chronic rejection, because interferon and ribavirin both have potent immunomodulatory properties, resulting in a broad range of immune-related disorders that include acute cellular rejection and chronic ductopenic rejection [12]. A 17% rate of chronic rejection was reported among 70 HCV liver transplant recipients treated with a pegylated, interferon-based regimen [13].

Even in Asia, in which LDLT predominates, the overall experience using living donor grafts for re-transplantation in patients with chronic allograft failure is limited. In a paper on liver re-transplantation from a large-volume center, only three patients received living donor-to-living donor re-transplantation in the series of first 1000 LDLTs. All three cases were performed for early graft failure [14]. LDLT-associated re-transplantation can be categorized into three types according to the sequences of graft types: living donor/living donor, living donor/deceased donor, and deceased donor/living donor. It also can be classified into two types according to re-transplantation timing: early (<6 months after primary transplant) and late (≥6 months after primary transplant). Living donor/deceased donor and living donor/living donor in the early period can be the most common types of LDLT-associated re-transplantation because all the vascular structures of the primary liver graft would be removed before the formation of adhesions, and new graft can be anastomosed to the recipient’s native structures. However, even in experienced, large-volume centers, late living donor/living donor re-transplantation is considered to be almost not feasible to perform due to heavy adhesion and distorted hilar structures that do not permit sufficient dissection [14].

The patient had sepsis accompanied by pneumonia before re-transplantation. The exact prevalence of sepsis in the emergency surgical population will vary by institution, but European data demonstrated that surgical patients were the most common category admitted to intensive care with sepsis and 47% suffered pneumonia [15]. Studies conducted in Italy [16] and Germany [17] showed sepsis mortality rates of 40% and 26% in post-re-transplantation, respectively. The recommendations of an expert panel commended by the Italian Association for the Study of the Liver (AISF) stated that a patient with any infectious event might be temporary suspended from the transplant list until complete resolution [18]. The patient was profoundly sick and received multi-system support on the intensive care unit that continued during his trip to the operating room. Mechanical ventilation had been instituted, together with invasive lines for both monitoring and the administration of drugs and fluid, having all resuscitative measures in place. Preoperative mechanical ventilation [19] and the MELD score [20] have been found to significantly affect outcome after re-transplantation. Emergency surgery may have a higher rate of complications after surgery. This is especially true for major surgery in critically ill patients. Even when surgery and anesthesia are straightforward, one in three high-risk patients develops serious medical complications in the days following surgery [21].

The rituximab might have caused the sepsis and pneumonia in the patient. However, patients with liver failure often develop sepsis as a result of the dysfunction of the defensive mechanisms against bacterial, viral, or fungal infections [22]. Infections in patients, even with cirrhosis, were reported to increase mortality fourfold [23]. Rituximab, a chimeric murine/human monoclonal antibody, reacts with the CD20 antigen, suppressing all stages of B lymphocyte differentiation except stem cells and long-lived plasma cells. However, the exact mechanism has not yet been elucidated [24]. The rapid effect of rituximab was reported in B cells, which were eliminated within 48–72 h of administration, with a single dose being sufficient for sustained suppression of B cells for several months in the post-transplant period [25]. In this case, rituximab was administered in a single dose of 600 mg (375 mg/body surface area) as a last viable treatment option for steroid-resistant liver allograft rejection, even if the effectiveness of treatment has not been verified. There was no plan for re-transplantation at the time. However, retrospectively, the drug was administered 60 days before re-transplantation. There was a report of a successful rescue therapy of a multi-drug-resistant liver allograft rejection with rituximab, alleging that the addition of rituximab might be a valuable option to overcome severe, multi-drug-resistant rejection, although a humoral nature of rejection is not proven by histology [26].

The eligibility for liver re-transplantation was determined by our transplant team on a case-by-case basis by a judgement that pneumonia and sepsis were difficult to improve in the state of liver failure. Although this re-transplant has not sufficiently been corroborated in many patients, at least it could be used as an example of extending the indication in patients who suffer from graft failure and are awaiting re-transplant. The major concern is what the selection criteria are; therefore, further refined studies need to be performed. In clinical practice, the selection of patients eligible for emergency LDLTs for re-transplantation to achieve the best outcome and survival should be evaluated by close integration with surgical expertise, clinical (age, performance status, MELD score) parameters, and careful patient monitoring on an individual basis. The only way to obtain meaningful data regarding this issue would be through well-designed prospective studies, using adequate severity-scoring systems to assess the impact of clinical variables on patients’ outcomes. It is important that critically ill patients with graft failure are enrolled into multicenter studies addressing these questions. Until such data are obtained, clinical judgement will still be the best tool in making decisions regarding the individual patients.

This patient’s operation may not have had the successful outcome that the patient’s family and medical staff had hoped for, and possible bad outcomes may have created feelings of guilt or regret for the family and medical staff. However, their values and beliefs were incorporated into a shared decision-making for this high-risk surgery. The same desperate need of a drowning person to clutch at a straw made it possible to perform re-transplantation. This ABO-incompatible LDLT was a last resort for the patient because blood-type-incompatible, deceased-donor liver transplantation has never been and still is not allowed in South Korea.

Last but not least, the results of this study suggest that, even in critically ill patients with liver graft failure complicated with sepsis and pneumonia, urgent re-transplantation may be life-saving. The good outcomes could be attributed to the following factors. First, the graft source was a young, living donor rather than a brain-dead donor, which played a big role in the decision to retransplant. Second, the patient’s condition was getting serious at the time of the LDLT, reflected by high MELD score. However, the heart function was maintained to some extent enough to withstand surgery. Third, the graft without significant hepatic steatosis or other parenchymal disease had no long ischemic time, and the graft implantation resulted in no derangements in the vascular inflow and outflow, which could be corroborated by the observation that the liver functions were maintained stably throughout hospital stay. Fourth, the patient had no immediate postoperative morbidity such as infection, rejection, and vascular or biliary complications, although biloma was detected at 3 months after surgery that could be a relatively stable period. Any immediate complication after surgery may tip the balance of patient recovery, especially in this critically ill patient with liver failure and sepsis.

In conclusion, long-term, satisfactory outcomes can be achieved in emergency, ABO-incompatible, adult-to-adult, living-donor liver re-transplantation for graft failure complicated by pneumonia-related sepsis in selected patients.

## Figures and Tables

**Figure 1 jcm-12-01110-f001:**
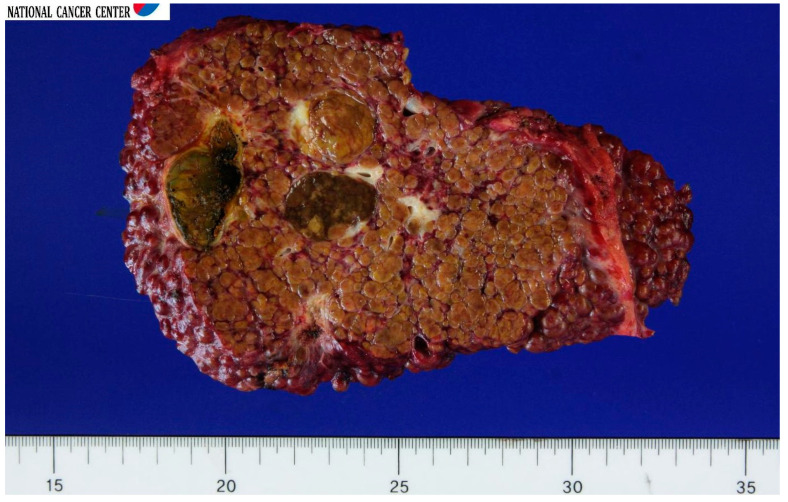
Gross specimen of the whole liver explant showing multiple hepatocellular carcinoma nodules and cirrhosis at the time of the first transplantation.

**Figure 2 jcm-12-01110-f002:**
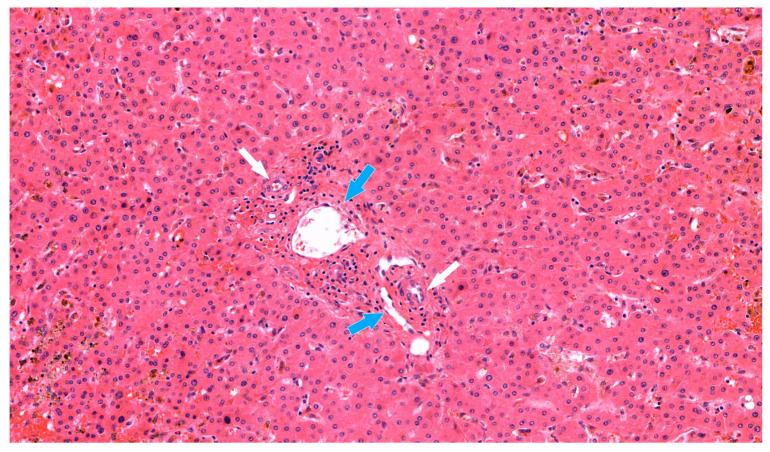
The liver allograft biopsy suggestive of chronic rejection (HE stain, 170×). In the portal tract, inflammatory cells were observed around hepatic artery (white arrow) and portal vein (blue arrow). Loss of interlobular bile duct was apparent with marked canalicular cholestasis.

**Figure 3 jcm-12-01110-f003:**
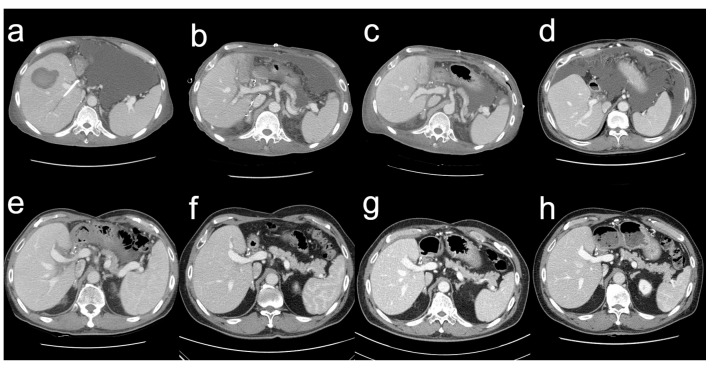
Series CT scans on pre-retransplant day 12 (**a**), postoperative day 10 (**b**), postoperative day 28 (**c**), postoperative month 3 (**d**), postoperative year 1 (**e**), postoperative year 3 (**f**), postoperative year 5 (**g**), and postoperative year 8 (**h**).

**Figure 4 jcm-12-01110-f004:**
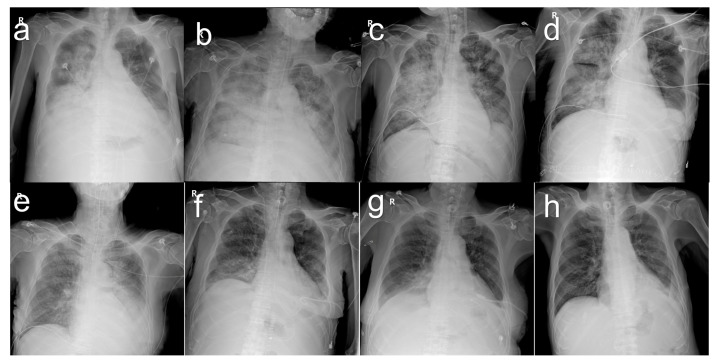
Series chest X-rays in the patient with pneumonia on pre-retransplant day 12 (**a**), preoperative day 0 (**b**), postoperative day 0 (**c**), postoperative day 4 (**d**), postoperative day 8 (**e**), postoperative day 16 (**f**), postoperative day 29 (**g**), and postoperative day 39 (**h**).

**Figure 5 jcm-12-01110-f005:**
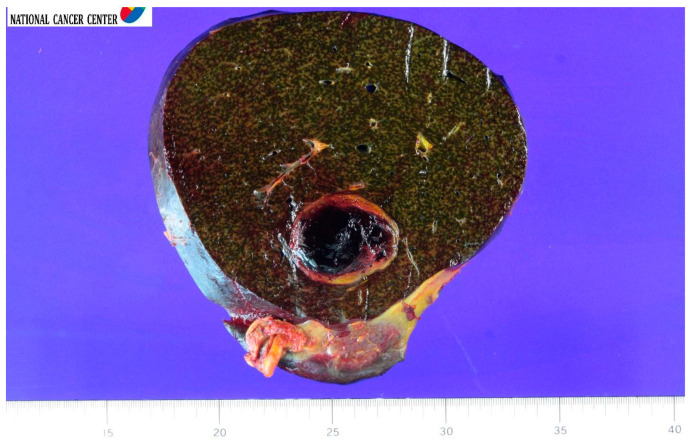
Gross specimen of the explanted right liver allograft demonstrating a green cholestatic cut surface with an intraparenchymal organizing hematoma measuring 4 cm in size at the time of re-transplantation.

## Data Availability

The data that support the findings of this study are available from the corresponding author, S.H.K., upon reasonable request.

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
