# Peer review of "Emergency, ABO-Incompatible Living Donor Liver Re-Transplantation for Graft Failure Complicated by Pneumonia-Associated Sepsis"

_jcm, 2023, doi:10.3390/jcm12031110_

Round 1

Reviewer 1 Report

Thank you for the opportunity to review this case report regarding an ABO incompatible living related liver re-transplant in a septic patient. Despite the odd grammatical or typing error, the author has painted a very lifelike story, and this is one of the more enjoyable case reports I have read in a while.

The discussion is a little too verbose and lacking in direction at times. I would prefer to see 3-4 points of discussion regarding the case report, each covered by its own paragraph and argued succinctly. For example, the section between lines 301 and 324 adds very little to the discussion in the way it has been written.

I would recommend being a little less negative regarding the physician opinions in this paper, perhaps by referring to the issues as concerns rather than a specific physician’s opinion. Also, I would be a little more humble in the first paragraph in the discussion when using hindsight to claim the most appropriate treatment.

I would suggest providing some results in between the first transplant and the 6 month mark that show good graft function, as the first graft was potentially small for size with a GRWR of 0.72 – and a potential cause or contributory factor to early graft failure.

Some of the glaring grammatical errors and typos include:

Line 224 – ‘door’ should be ‘donor’

Words that don’t need to be pleural, including but not limited to line 30 “conditions” and “functions”, line 44 “patients”, and line 267 “patients”.

I would recommend editing by an English editor to improve the language and help correct these errors.

Author Response

Reviewer 1’s Comments to the Author:
Thank you for the opportunity to review this case report regarding an ABO incompatible living related liver re-transplant in a septic patient. Despite the odd grammatical or typing error, the author has painted a very lifelike story, and this is one of the more enjoyable case reports I have read in a while.

The responses(s): Many thanks to your encouraging comments. The odd grammatical or typing error were corrected.

The discussion is a little too verbose and lacking in direction at times. I would prefer to see 3-4 points of discussion regarding the case report, each covered by its own paragraph and argued succinctly. For example, the section between lines 301 and 324 adds very little to the discussion in the way it has been written.

The responses(s): The discussion was described on 7 important points and each was covered by its own paragraph and argued succinctly. And, according to the reviewer 2’s comment, in the lines 301-306, we expanded on the ethical implications. And, the lines 307-324 was revised to be a single paragraph.

I would recommend being a little less negative regarding the physician opinions in this paper, perhaps by referring to the issues as concerns rather than a specific physician’s opinion. Also, I would be a little more humble in the first paragraph in the discussion when using hindsight to claim the most appropriate treatment.

The responses(s): Many thanks to your thoughtful comments. In the revised version, the word ‘clearly’ was changed into ‘humbly’.

I would suggest providing some results in between the first transplant and the 6 month mark that show good graft function, as the first graft was potentially small for size with a GRWR of 0.72 – and a potential cause or contributory factor to early graft failure.

The responses(s): The patient showed normal liver function up to 6 months after LDLT. At 3 months after LDLT, the serum total bilirubin level, AST level and ALT level were 0.6 mg/dl, 20 U/L, and 18 U/L, respectively. At 5 months after LDLT, the serum total bilirubin level, AST level and ALT level were 0.4 mg/dl, 32 U/L, and 14 U/L, respectively.

Some of the glaring grammatical errors and typos include:

Line 224 – ‘door’ should be ‘donor’

Words that don’t need to be pleural, including but not limited to line 30 “conditions” and “functions”, line 44 “patients”, and line 267 “patients”.

I would recommend editing by an English editor to improve the language and help correct these errors.

The responses(s): The grammatical or typing error were corrected following your comments.

Reviewer 2 Report

Title

-        State explicitly that this is a case report

Abstract

-        State explicitly that the nephew donor was ABO incompatible

-        Include the blood types of the patient, 1st donor and 2nd donor

-        Remove the section ‘Materials and Methods’ as this is a case report, not a research study

Case presentation

-        Line 154-155 – Needs to be more objective; ‘as if looking for a ray of hope’ is a particularly unnecessary diversion

Discussion

-        Lines 224-225 – Do you have this statistic for Korea (or Asia)? I believe the rates of viral hepatitis are quite different.

-        Lines 232-324 – What population is this statistic from? How does this compare to typical rates of chronic rejection?

-        Lines 240-241 – This terminology may be confusing. Instead of saying using ‘to’, maybe say living donor/living donor, living donor/deceased donor, etc. or living donor followed by living donor, etc.

-        Lines 301-306 – Expand on the ethical implications. Would this scenario be possible with a deceased donor given how unwell the recipient was and the incompatibility of the graft?

Grammar

-        Line 18 – the ‘s’ in ‘Serum’ should be lower case

-        Line 64 – report the abbreviated units for platelets (i.e., /μL)

-        Lines 307-324 – need to be a single paragraph

Author Response

Reviewer 2’s Comments to the Author:

Title

-        State explicitly that this is a case report

 The responses(s): I state clearly that this is a case report, following your opinion.

Abstract

-        State explicitly that the nephew donor was ABO incompatible

The responses(s): I stated clearly that the nephew donor was ABO incompatible, following your opinion.

-        Include the blood types of the patient, 1st donor and 2nd donor

The responses(s): I Included the blood types of the patient, 1st donor and 2nd donor.

-        Remove the section ‘Materials and Methods’ as this is a case report, not a research study

The responses(s): I removed the section ‘Materials and Methods’ as this is a case report, not a research study

Case presentation

-        Line 154-155 – Needs to be more objective; ‘as if looking for a ray of hope’ is a particularly unnecessary diversion

The responses(s): I removed the phrase ‘as if looking for a ray of hope’.

Discussion

-        Lines 224-225 – Do you have this statistic for Korea (or Asia)? I believe the rates of viral hepatitis are quite different.

   The responses(s): We don’t have the statistics for Korea because the prevalence of HCV is far less than that of HBV.

-        Lines 232-324 – What population is this statistic from? How does this compare to typical rates of chronic rejection?

The responses(s): Patients with postLT recurrent HCV undergoing antiviral treatment were identified through the Recanati-Miller Transplantation Institute database, Mount Sinai School of Medicine, New York, NY. The reference article (13) reported that a 17% rate of chronic rejection was reported among 70 HCV liver transplant recipients treated with a pegylated interferon-based regimen.

In the case, the patient complained of abdominal distension at 6 months after LDLT. The serum total bilirubin level was 2.7 mg/dL and the serum level of HCV RNA had increased to 6,890,807 IU/mL. So, the patient was diagnosed as postLT recurrent HCV, and Pegylated interferons was started in combination with ribavirin. This treatment may have played a role in causing chronic rejection because interferon and ribavirin have both potent immunomodulatory properties resulting in a broad range of immune-related disorders including acute cellular rejection and chronic ductopenic rejection.(12)

-        Lines 240-241 – This terminology may be confusing. Instead of saying using ‘to’, maybe say living donor/living donor, living donor/deceased donor, etc. or living donor followed by living donor, etc.

The responses(s): The terminology was changed into such as living donor/living donor, living donor/deceased donor.

-        Lines 301-306 – Expand on the ethical implications. Would this scenario be possible with a deceased donor given how unwell the recipient was and the incompatibility of the graft?

The responses(s): Deceased donor liver grafts are a rare and valuable public resource with limited supply and are subject to an equitable allocation system. However, living donor liver grafts are an exclusive gift for just one patient based on the close relationship between donor and recipient. This was clearly emphasized in line 313-314 as such that the graft source was a young living donor rather than a brain-dead donor, which played a big role in the decision to retransplant.

Blood type incompatible deceased donor liver transplantation has not been allowed in Korea until now, and is still not allowed. So, without the ABO-incompatible LDLT, the good outcome would have been impossible given the deteriorating patient’s condition.  

Grammar

-        Line 18 – the ‘s’ in ‘Serum’ should be lower case

-        Line 64 – report the abbreviated units for platelets (i.e., /μL)

-        Lines 307-324 – need to be a single paragraph

The responses(s): The grammar and spelling were corrected appropriately.

Round 2

Reviewer 2 Report

No additional comments